# Radiation Hardness Study of Single-Photon Avalanche Diode for Space and High Energy Physics Applications

**DOI:** 10.3390/s22082919

**Published:** 2022-04-11

**Authors:** Ming-Lo Wu, Emanuele Ripiccini, Ekin Kizilkan, Francesco Gramuglia, Pouyan Keshavarzian, Carlo Alberto Fenoglio, Kazuhiro Morimoto, Edoardo Charbon

**Affiliations:** 1AQUA Laboratory, École Polytechnique Fédérale de Lausanne (EPFL), 2002 Neuchâtel, Switzerland; emanuele.ripiccini@epfl.ch (E.R.); ekin.kizilkan@epfl.ch (E.K.); francesco.gramuglia@epfl.ch (F.G.); pouyan.keshavarzian@epfl.ch (P.K.); carlo.fenoglio@epfl.ch (C.A.F.); 2Canon Inc., Kawasaki 212-8602, Japan; morimoto.kazuhiro@mail.canon

**Keywords:** proton irradiation, radiation damage, radiation-tolerant, single-photon avalanche diode, SPAD, space application

## Abstract

The radiation hardness of 180 nm complementary metal–oxide–semiconductor (CMOS) and 55 nm bipolar–CMOS–double-diffused MOS single-photon avalanche diodes (SPADs) is studied using 10 MeV and 100 MeV protons up to a displacement damage dose of 1 PeV/g. It is found that the dark count rate (DCR) levels are dependent on the number and the type of defects created. A new stepwise increase in the DCR is presented. Afterpulsing was found to be a significant contributor to the observed DCR increase. A new model for DCR increase prediction is proposed considering afterpulsing. Most of the samples under test retain reasonable DCR levels after irradiation, showing high tolerance to ionizing and displacement damage caused by protons. Following irradiation, self-healing was observed at room temperature. Furthermore, high-temperature annealing shows potential for accelerating recovery. Overall, the results show the suitability of SPADs as optical detectors for long-term space missions or as detectors for high-energy particles.

## 1. Introduction

A large number of high-energy particles such as protons, electrons, and other heavy ions originating from the Sun and other celestial bodies outside of our solar system can impinge on the devices we send into space [1], whereas on Earth, charged particles are mostly deflected by the magnetic field and the atmosphere. To construct a system suitable for space applications, all the components should undergo a series of tests before their operation [2]. Radiation hardness testing is critical to understanding the risks related to the operation of electrical components in a radiative environment. It has been performed on the transistor level as well as on more complex components such as memories and field-programmable gate arrays (FPGAs) [3]. Radiation effects are also studied on silicon-based sensors such as photodiodes and charge-coupled devices (CCDs), as well as III–V-material-based photonic devices such as ring resonators, and lasers [4,5,6,7,8].

Single-photon avalanche diodes (SPADs) allow photon counting and single-photon resolution imaging when used in an array. The capability of photon counting makes SPADs the detector of choice for applications in which conventional photodiodes and CCDs cannot be used. In addition, the complementary metal–oxide–semiconductor (CMOS) compatibility of SPADs can also take advantage of high-speed electronics in close proximity for fast parallel processing. A well-designed SPAD also shows remarkably low noise and good timing performance. An array of SPADs can be used as a camera for intensity imaging with a high dynamic range [9,10,11,12,13]. The excellent timing performance, on the order of tens of picoseconds, makes them also powerful for 3D imaging or ranging applications, lidar [14,15,16], fluorescence lifetime imaging microscopy [17,18,19], Raman spectroscopy [20,21], quantum key distribution [22], and quantum random number generation [23]. A recent study also shows the possibility of using SPADs for direct high-energy particle detection taking advantage of the single event effect [24]. All the applications mentioned above can benefit the growing space exploration and space-based physics studies. For SPADs, it has been shown that the dark count rate (DCR) increases due to displacement damage caused by protons and ionizing damage caused by X-rays, alpha-particles, and neutrons [25,26,27,28,29,30]. To sustain long-term operation in harsh environments, such as space, and to evaluate the plausibility of SPADs replacing photodiodes and CCDs in more applications, it is important to study the radiation hardness of SPAD detectors. In this paper, we study the effect of proton or hydrogen nuclei radiation, which is one of the main radiation sources given by cosmic rays, on all the figures of merit of SPADs. Our study covers DCR, afterpulsing probability (APP), timing jitter, and photodetection probability (PDP). We report a SPAD DCR increase that is several orders of magnitude lower after delivering a higher radiation dose compared to existing literature. We were also able to study the influence of such high radiation on timing jitter with picosecond-level precision thanks to the recently developed accurate-timing pixel. In this work, we also proposed a new model of DCR increase based on the newly observed APP degradation. We then further investigated the underlying device physics that causes all these changes in the listed figures of merit.

The paper comprises four sections. Section 2 presents the devices under test and the method used to characterize the irradiated samples. The results, the simulations, and the analysis to support the outcome are shown in Section 3. Section 4 concludes the paper.

## 2. Materials and Methods

In this work, we characterize three types of SPADs. The first device under test (DUT) is a low noise SPAD fabricated in a 55 nm bipolar–CMOS–double-diffused MOS (BCD) process presented in [31,32]. SPADs with different junction depths and active radii in the range of 1.6 to 4.6 μm were studied for each device. The second DUT comprises larger SPADs with active radii of 8.8 μm and 21.4 μm and an accurate timing readout designed in a 180 nm CMOS process presented in [33]. The last DUT is the 180 nm CMOS megapixel SPAD camera presented in [9].

We irradiated the samples using the Proton Irradiation Facility at Paul Scherrer Institute (PSI, Villigen, Switzerland). Mono-energetic beams with tunable flux were used to characterize the behavior of SPADs under radiation. The protons were collimated into a 30 mm square area with less than 10% non-uniformity. The SPADs were placed with their front side towards the incident beam. The dose rate, which corresponds to the proton flux (proton/cm^2^·s), was adjusted to reach a certain dose step within a reasonable time. All samples were unbiased when irradiated to avoid self-healing due to SPAD firing so that the damage caused by protons could be observed independently [25]. All DUTs were prepared in pairs so that the effects of both 100 MeV and 10 MeV protons can be compared.

Protons cause both ionizing damage and displacement damage, in which the total delivered dose can be quantified by the total ionizing dose (TID) and displacement damage dose (DDD), respectively. It was shown that damage caused by TID is transient, as recovery from ionizing damages can occur at room temperature [34]. On the contrary, damages caused by DDD result in vacancy–interstitial defects, from which recovery requires higher energy or higher temperature annealing [35]. Proton energy of 100 MeV and 10 MeV were used to study the effect of different incoming particle energies. The proton energies were obtained by adding degraders to a 230 MeV accelerator source, resulting in actual delivered energy of 101.34 MeV and 10.29 MeV, respectively.

DDD steps were set to be the same for both 100 MeV and 10 MeV protons. Note that due to less interaction of high energy protons, or less linear energy transfer to the silicon lattice, the sample irradiated with 100 MeV protons received 31.5 krad TID while the sample irradiated with 10 MeV protons received 58.9 krad TID when both samples reached a DDD of 1 PeV/g. The corresponding TID and flux used for each dose step are shown in Table 1.

To characterize all the samples at each radiation step, the 55 nm SPADs were wire-bonded to a printed circuit board so that the SPADs could be passively quenched by a 150 kΩ resistor. The SPAD pulses were digitized by an array of LP339 comparators and then accumulated using an XEM7360 FPGA. The number of pulses at each second was measured, and the DCR is defined as the average counts per second (cps) of a 1-minute measurement in darkness. 

The DCR of the megapixel SPAD arrays was measured by taking images in a dark environment with known exposure time. All DCR measurements were performed within 10 min after every dose step was reached. This was to avoid room-temperature annealing and to capture the DCR at its highest level.

After the last dose of proton radiation, the samples were kept at room temperature and the DCR of each sample was measured several times to observe the DCR change over four weeks. After 30 days, the samples were annealed using a universal oven at a high temperature of up to 160 °C to remove residual defects.

## 3. Results

In this section, we show the characterization result of the samples before and after irradiation and after annealing. For the DCR measurement, the DCR level was measured at every dose step. For the APP, PDP, and jitter, the results compare only the performance before and after the last dose of irradiation due to the long measurement time requirement.

### 3.1. Breakdown Voltage

At room temperature, the breakdown voltages of the 55 nm BCD SPADs are 32 V and 19.5 V for the deep [31] and shallow junctions [32], respectively. The breakdown voltages of the 180 nm CMOS accurate timing SPAD [33] and the megapixel SPAD camera [9] are 22 V and 22.8 V, respectively. For a passive-quenched SPAD, the breakdown voltage can be calculated by the operating voltage minus the output pulse voltage seen on the anode of the SPAD. For SPADs with front-end circuitry, the breakdown voltage of the SPAD array can be obtained by measuring the counts from the detector output over different operating voltages under a uniformly illuminating condition. The breakdown voltage can then be defined as the intercept on the operating voltage axis [36]. For both 55 nm and 180 nm SPADs, we observed no change in breakdown voltage after irradiation in any of the DUTs. This is consistent with all previous studies [37,38], showing that there is no major doping profile change after reaching a DDD of 1 PeV/g.

### 3.2. Dark Count Rate

DCR is one of the critical figures of merit of SPADs for many applications as it represents the noise level. For imagers, higher DCR will result in a lower signal-to-noise ratio or image quality. For single-photon counting applications such as quantum key distribution, the single-photon detector noise contributes to the error rate of key transmission. If the DCR exceeds a certain threshold, it can cause the quantum communication protocol to fail [39].

Next, we show the effect of radiation damage on DCR. In Figure 1, three of the characterized 55 nm SPADs are shown. All DCR measurements were performed at five different excess bias voltages (Vex). The lower and upper horizontal axes represent the cumulative DDD and TID, respectively. Three different behaviors of DCR under increasing cumulative dose can be observed. Figure 1a shows a SPAD with continuous growth in DCR as a function of the cumulative dose. Figure 1b shows how SPAD DCR peaks at a certain dose and then starts to drop during irradiation. This behavior is the most counter-intuitive one, but in fact, most SPADs exhibit this trend at some point during irradiation. This kind of short-term relaxation is also reported in [27]. Figure 1c shows one SPAD with a stepwise increase in DCR. To the best of our knowledge, this behavior has not been reported elsewhere to date. All these characteristics are observed in multiple SPADs. Most DUTs show a mixture of the aforementioned behaviors.

We hypothesize that this stepwise increase is most likely related to the size of the SPADs that were tested. To cause DCR to increase, the high-energy particle has to create damage within or close to the photocollector region of the SPAD. If the SPAD is small enough that no defect was created between dose steps at that region, no DCR degradation will occur. There is a certain probability that no interaction will occur between the protons and the atoms in the SPAD photocollector region. Therefore, we also observed SPADs with no increase in DCR after the DDD reaches 1 PeV/g. All of the 55 nm SPADs with active radii smaller than 2 μm show either stepwise increase or no DCR increase at all.

This hypothesis is consistent with the results obtained for the larger 180 nm SPADs we tested. For the DUTs with an 8.8 μm active area radius, we irradiated the sample with 100 MeV protons. The average DCR increased from approximately 100 cps before irradiation to 36 kcps at a DDD of 300 TeV/g. For the DUTs with a 21.4 μm active radius, we irradiated the sample with 10 MeV protons. The average DCR increased from around 1 kcps before irradiation to 200 kcps at a DDD of 200 TeV/g. It is evident that larger SPADs undergo worse degradation.

In Figure 2, the evolution of the cumulative DCR distribution of the megapixel SPAD arrays for increasing DDD is shown. The value in the legend represents the DDD in TeV/g, where DDD = 0 represents the DCR before irradiation. Both arrays start with similar mean and median DCR. With this large amount of SPADs in the array, we can compare the effects of 100 MeV and 10 MeV protons more precisely.

We can see that the DUT exposed to 10 MeV protons suffer from a more significant DCR increase compared to the one under 100 MeV at the same DDD. This is because, despite the fact that both samples received the same total dose, the damage created at the depth of the photocollector region is different. This was verified using Transport of Ions in Matter (TRIM) simulation, which allows the estimation of transport of atoms due to ion collision. The simulation result is shown in Figure 3. In this figure, the protons enter silicon from the left side and interact with silicon. The trajectories of the protons are shown in white. We can see how 10 MeV protons interact with silicon more while the 100 MeV protons penetrate the silicon chip without much scattering. The orange and red curves show the number of vacancies per proton per angstrom in depth created by 100 MeV and 10 MeV protons, respectively. The estimated displacement damages at the depth of the photocollector region given by 100 MeV and 10 MeV are 7 × 10^−7^ and 7 × 10^−6^ vacancies/proton·Å, respectively. With the different total fluences delivered that are shown in Table 1, the estimated defect counts within the active region of each pixel are 43 and 140 for 100 MeV and 10 MeV protons, respectively. The ratio of the defect counts of the two DUTs, roughly 1:3, matches the ratio of the final mean DCR for both DUTs, which is shown in Figure 4. In the case of the DUT irradiated with 100 MeV protons shown in Figure 4a, the exposure was discontinuous and separated into three days due to the beamtime arrangement. The drop in mean DCR between doses on different days is evidence of room-temperature annealing.

A mixture of all the three cases shown in Figure 1 can also be found in the SPAD pixels of the megapixel camera. This shows that the three types of DCR evolution explained in Figure 1 are independent of SPAD structure, doping profile, and the technology node.

Finally, we can see how hot pixels, defined as pixels with a DCR significantly higher than the median DCR, are evolving in Figure 2. Before irradiation, there are two clear knees at 85% and 99% cumulative probability where the pixels show one or more orders of magnitude higher DCR than the rest of the pixels. This is commonly seen in large format SPAD arrays [9,17,40]. As the DDD increases, we can see that the two knees shift towards the left, indicating the number of hot pixels is increasing progressively. The difference in DCR levels is likely to be caused by different types of defects within the silicon bulk or silicon–oxide interface. During fabrication or receiving radiation, interface defects, oxygen–vacancy, phosphorus–vacancy, boron–vacancy, and more different complexes can emerge [41,42]. All these defects result in deep-level traps. Depending on the type of defect, the electron and hole traps will emerge at different energy levels within the bandgap of silicon. From the Shockley–Read–Hall model and Fermi–Dirac statistics, we know that the generation and recombination rate of charged particles is dependent on the position of the trap energy within the bandgap. From the difference in DCR levels, we can speculate that they are originating from the difference in trap energy levels. The knees are also less apparent at a higher cumulative dose, which can be the indication of all types of defects being formed uniformly within the array.

#### 3.2.1. Deep-Level Trap Activation Energy

The DCR level can be lowered after radiation by cooling down the SPAD. Defects created by high-energy protons can act as deep-level traps which capture electrons and holes. At higher temperatures, these trapped charges have higher chances to escape from the trap and trigger an avalanche if they enter the high-field region. We can see in Figure 5a the temperature dependency of the mean DCR of the megapixel SPAD camera before and after irradiation. By taking binary frames, the afterpulsing component is suppressed as we only capture the primary dark count. We can then determine the activation energy of these traps by using the Arrhenius law with DCR ∝ exp(−*E_act_*/*k_b_T*), where *E_act_* is the activation energy, *k_b_* is the Boltzmann constant, and *T* is temperature [43]. We find that the median *E_act_* changes from 1.0 eV before irradiation to 0.5 eV after irradiation. This again is evidence of emerging deep-level traps introduced by radiation damage.

From Figure 5b,c, we can observe the *E_act_* of the pixels before and after 100 MeV and 10 MeV irradiation. Before irradiation, we can identify two groups of pixels, A and B, with *E_act_* of ~1.1eV and ~0.8 eV, respectively. These two groups correspond to the two groups of SPADs shown in Figure 2, where the two knees are presented. This is proof of the correlation between the trap energy and the DCR level. Before irradiation, the DCR of the pixels is determined by the type of the existing defects from the fabrication process.

We can also see how the *E_act_* shifts from silicon bandgap (1.1 eV) towards half bandgap after irradiation. The wider spread in *E_act_* explains why the knees in Figure 2 become less apparent at higher cumulative doses. We can identify several distinctive activation energies from the two figures. These activation energies, 0.16 eV, 0.38 eV, 0.44 eV, and 0.55 eV, coincide with the energy levels of oxygen–vacancy complex, divacancy, phosphorus–vacancy, and mid bandgap, respectively. These energy levels can also be seen in spectroscopy studies in CMOS image sensors and SPADs, as well as photoconductivity measurement of electron-irradiated silicon [41,42,43,44,45]. The spread in activation energies can be caused by trap-assisted tunneling and the Poole–Frenkel effect, which lowers the energy required for carrier transportation [44,45,46]. Depending on the electric field strength where the defect is created spatially, the observed active energies will differ from the trap energy level. The two DUTs have similar median activation energy at 0.5 eV, showing that the DCR levels are more dependent on the densities of the defects that have been created after irradiation.

#### 3.2.2. Afterpulsing Probability

We noticed that in several 55 nm SPAD pairs, the DUT exposed to 10 MeV protons increases by several orders of magnitude in DCR compared to its counterpart exposed to 100 MeV protons. This does not agree with the aforementioned defect counts that we obtained from our simulations. We observed that strong afterpulsing dominates the dark count statistics in these SPADs.

Afterpulses can occur when charge carriers, which were trapped from previous avalanches, are released. Depending on the SPAD breakdown probability, these released carriers may ignite another avalanche, thereby contributing to the overall DCR. Afterpulsing can be measured by putting the SPADs in a dark environment and measuring the inter-arrival time of noise pulses. For a Poisson process, the inter-arrival time should follow an exponential distribution. Since the primary dark count is a Poisson process, the inter-arrival time histogram should present a single exponential decay in the absence of afterpulse, whereas the contribution above the exponential fit is defined as afterpulsing [47]. Here, we define APP as the number of pulses above the exponential fit divided by the total number of pulses. The definition is expressed in Equation (1).
(1)$$\mathrm{APP}\left(\%\right)=\frac{\mathrm{Afterpulses}}{\mathrm{Primary}\;\mathrm{dark}\;\mathrm{counts}+\mathrm{Afterpulses}}=\frac{\mathrm{Afterpulses}}{\mathrm{Dark}\;\mathrm{counts}}\times 100\%$$

In Figure 6a, a SPAD with negligible afterpulsing before radiation is shown. The majority of the DUTs showed APP < 1% before radiation damage. In Figure 6b, afterpulsing characterization was performed on the same SPAD with different excess biases after radiation damage.

It can be seen clearly that there is a dramatic increase in afterpulsing. The afterpulsing becomes more and more dominant in the contribution of DCR at higher excess bias. This can be seen in Figure 7a, where thirteen measured SPADs show the same behavior. A SPAD that has a 50% afterpulsing probability indicates every signal pulse or a primary dark count will be followed by an afterpulse. This might not be a problem for imagers operating at a low frame rate as every binary frame only registers the first pulse, or the pulse that origins from the true signal. This is also the reason why the DCR is proportional to the defect count for the 180 nm megapixel camera. However, this failure mechanism can be an issue for communication applications such as QKD, in which afterpulsing can cause a loss in the efficiency of the setup [48,49]. Afterpulsing is also devastating for quantum random number generators, as it introduces correlation between pulses [50].

We can also see in Figure 7a that most SPADs irradiated with 10 MeV protons show higher afterpulsing probability compared to the SPADs irradiated with 100 MeV protons. This can be due to the fact that SPADs irradiated with 10 MeV have more defects within the depletion region, as explained at the beginning of this section. From [51], we can model the time-dependent afterpulsing count as follows:(2)$$P_{ap}\left(t\right)=\overset{N}{\underset{i=1}{\sum }}A_{i}\frac{1}{\tau_{i}}e^{\frac{-t}{\tau_{i}}}$$
where *N* is the total number of deep-level traps, *A_i_* is an exponential prefactor constant, and *τ_i_* is the lifetime of the *i*-th trap. As the trap lifetime is temperature-dependent, we also study the relation between APP and temperature. We measured the APP of three SPADs at different temperatures. The voltage is tuned at each temperature step to operate the tested SPAD at 3 Vex. The result is shown in Figure 7b. We can see that due to the longer lifetime τ at lower temperatures, the APP increases in an exponential trend for the three tested SPADs. When the lifetime of the trap is longer, there is a higher chance to trigger an avalanche during the recharge phase of a SPAD, while at higher temperatures, the trapped carriers might be released before the SPAD recharges.

We can see in Figure 6b that afterpulses create a multi-exponential component on top of the single exponential Poisson statistic with a trap lifetime up to tens of microseconds. This agrees not only with the time-dependent afterpulsing model in Equation (2) but also with the aforementioned different trap energy levels seen in the cumulative DCR distribution of the 180 nm SPAD camera. It is thought that this is further evidence of several defects with different trap energy levels created during irradiation.

In most of the existing literature, the DCR increase is modeled with a linear fit to the cumulative dose. However, we find this not to be accurate. The DCR should be modeled as a function of not only DDD but also APP after radiation. From the unitless Equation (1), we can rewrite the primary dark count and afterpulse count into DCR and afterpulsing rate in cps and find the following:(3)$$1-\mathrm{APP}=\frac{\mathrm{Primary}\;\mathrm{DCR}\;\left[\mathrm{cps}\right]}{\mathrm{DCR}\;\left[\mathrm{cps}\right]}$$

We can then write the DCR under radiation as follows:(4)$$\mathrm{DCR}\left(\mathrm{DDD},\mathrm{APP}\right)=\frac{DCR_{0}+\left(K_{d}\times V_{dep}\times \;\mathrm{DDD}\right)}{1-\mathrm{APP}}=\frac{{\mathrm{Primary}\;\mathrm{DCR}}_{\mathrm{rad}}}{1-\mathrm{APP}}\;[\mathrm{cps}]$$
where *DCR_0_* (in cps) is the original DCR before radiation assuming no afterpulsing. The second term in the numerator represents the linear increase in the primary dark count or the linear increase in the leakage current of a diode due to the increase in the number of defects. *K_d_* (in carriers per cm^3^ per MeV/g) is a constant called damage factor that describes the rate of damage given by the radiation source, and *V_dep_* (in cm^3^) is the depleted volume [27,52]. In other words, the primary dark count will indeed be linearly proportional to the DDD. The numerator represents the total primary DCR_rad_ after receiving a certain dose of radiation. The denominator is the scaling term considering afterpulses caused by primary dark counts, which is derived in Equation (3).

In Figure 8, we compare the proposed model shown in Equation (4) with the known linear model. We used the megapixel SPAD camera to acquire the mean *K_d_*. The *K_d_* values of 100 MeV and 10 MeV protons are 6.7 × 10^4^ and 2.2 × 10^5^, respectively. This is similar to what has been shown in [52]. We then apply the acquired *K_d_* and the APP to find the DCR. In Figure 8a, five measured SPADs are shown. We can see that the DCR values acquired with afterpulsing correction have much better accuracy for the SPADs with high DCR increase. However, the linear model, not considering the influence of afterpulsing, underestimates the DCR level.

It is difficult to conclude how APP will increase with cumulative dose since the afterpulsing characterization was only done before irradiation and after the last dose of irradiation. However, if we assume the ratio of afterpulses and primary dark counts increases linearly with cumulative dose, we can expect the DCR to increase in a quadratic trend. In Figure 8b, an example of DCR prediction is shown. In this example, we used the known *K_d_* and the assumption of APP = 90% at 1 PeV/g. We can again see that the model considering afterpulsing gives a better DCR increase prediction.

### 3.3. Dark Count Rate Random Telegraph Signal

We observed the phenomenon of dark count rate random telegraph signal (RTS) in damaged SPADs. RTS was not detected in any of the DUTs before irradiation. RTS has been reported several times in literature [53,54] and in several DUTs in this work. An example of two 55 nm SPADs showing RTS behavior is shown in Figure 9. Since the 55 nm SPADs are passively quenched without any integrated readout circuit, we can confirm that this random telegraph noise is coming from the SPADs themselves. RTS can be explained by the bi-stable or multi-stable defect within the silicon bulk, resulting in a random change of two or more DCR levels.

### 3.4. Photodetection Probability

The PDP of a SPAD represents the efficiency of detecting a photon and generating an associated electrical pulse. This is highly dependent on the quantum efficiency and breakdown probability. All these factors are dependent on the structure and doping profile of the SPAD. As with the breakdown voltage, there is no observable change in the PDP in any of the tested DUTs. This again shows that the structure and doping profile of the SPAD is unaffected by radiation, similar to what has been reported in [29].

### 3.5. Jitter

Timing jitter expresses the precision of a SPAD registering the arrival time of an impinging photon. It is defined as the timing uncertainty built up from the time a photon impinges until it is registered as an electrical signal. Jitter is dependent on the intrinsic time for a photoelectron to be multiplied in the high-field avalanche region, or the time needed for a photoelectron to diffuse from a low-field region to the multiplication region.

To study if radiation damage influences the performance of timing jitter, we used the 180 nm SPAD designed for precise single-photon counting presented in [33]. The sensor was characterized before radiation exposure and after receiving 300 TeV/g DDD. The result of jitter measurement at the wavelength of 780 nm is shown in Figure 10, where no jitter degradation was observed. The error bar represents the standard deviation between repeated measurements. This is as predicted because a few hundred vacancies within the SPAD should not affect charge diffusion and the process of triggering an avalanche. The slight difference between measurements can be caused by sample alignment with the experimental setup which creates an error at the picosecond level.

### 3.6. Annealing

Annealing has been applied not only in SPADs but also in photomultiplier tubes and charge-coupled devices to lower the effect of radiation damage [54,55,56]. In this work, we present the results of SPADs recovering under both room-temperature and high-temperature annealing.

#### 3.6.1. Room-Temperature Annealing

As already seen in Figure 1 and Figure 4a, room-temperature annealing can occur during irradiation or between days of discontinuous exposure. This is similar to the rapid self-annealing discussed in [25]. After the last dose of exposure, we kept all the DUTs under room temperature to observe this phenomenon. In Figure 11, we can see the DCR of the megapixel cameras drops rapidly within three days after the last dose of exposure, and then the decreasing rate slows down after four weeks. This is similar behavior to the one shown in [37]. It is clear how the sample that was exposed to 10 MeV has a larger DCR drop. The median DCR recovered percentages are 20% for the 100 MeV and 40% for the 10 MeV DUT. This can be due to the higher TID received by the 10 MeV sample since recovery from displacement damages is less likely under room temperature. It can also be seen that the mean DCR ratio of the two DUTs at the last measured day is still similar to the defect count ratio acquired from the TRIM simulation.

We find this recovery trend very similar to the annealing of metal–oxide–semiconductor (MOS) devices under X-ray and cobalt-60 radiation, which are forms of ionizing radiation [57]. In such irradiation, it has been found that the threshold voltage shift recovers over weeks in a logarithmic trend [58]. Similarly, our DUTs show a logarithmic-like DCR recovery. We hypothesize this to be the transient response caused by ionizing damage at the oxide interface. Due to cumulative TID, charges can be trapped at the surface of the SPAD, where oxide is used for passivation, or at the isolation–silicon interface, where oxide is used in trenches to prevent electric leakage between adjacent devices. The reduction in SPAD DCR can originate from the neutralization of oxide-trapped charges.

#### 3.6.2. High-Temperature Annealing

After room-temperature annealing, the DUTs were put in the oven for high-temperature annealing. At each temperature step, the DUTs were kept at the same annealing temperature for one hour and then cooled down slowly to room temperature for DCR characterization. The temperature range used was 100 °C to 160 °C in 20 °C steps.

Figure 12 shows the result of high-temperature annealing of both samples irradiated with 100 MeV and 10 MeV protons. Although the DCR drop can be caused by an acceleration of ionizing damage recovery due to higher temperature, we find the trend very similar to what was shown in [54]. The only difference is that our DUTs are still far from full recovery, whereas in [54], the SPADs are almost fully annealed, i.e., return to the DCR level before irradiation. The behavior of DCR having a larger drop between 120 °C and 140 °C is similar to the profile of defect concentration versus annealing temperature of a phosphorus–vacancy complex [41]. The unannealed fraction in our DUTs can originate from defect complexes that require higher temperatures to anneal. These defects can be oxide-vacancy complexes, arsenic–vacancy complexes, or divacancies [41].

Finally, we compare the activation energy of the 180 nm megapixel SPAD camera after irradiation and after annealing. We can see in Figure 13 that the average activation energy shifts uniformly towards the bandgap 1.1 eV. The mean activation energy shifts from 0.5 eV to 0.7 eV. We can still see a distinctive population around 0.4 eV for both DUTs, indicating some phosphorus–vacancy complexes and divacancies remain unannealed.

## 4. Discussion

We have investigated the effect of different energy proton irradiation on several SPAD systems designed in 55 nm BCD and 180 nm CMOS technology. SPADs irradiated with lower energy protons have larger and faster DCR increases compared to those exposed to higher energy protons. This is because lower energy protons interact with the silicon lattice more effectively, creating both a larger number of defects and ionized charges.

We discovered that severe afterpulsing can be a consequence of proton radiation damage. Moreover, the effect of afterpulsing is more pronounced when SPADs are operating in a more favorable regime, i.e., higher Vex for higher PDP and lower temperature for lower DCR. This problem is not addressed in most articles found in the literature regarding radiation damage in SPADs. However, we argue that this failure mechanism should always be considered, as afterpulsing can affect applications such as quantum communication and quantum random number generation. We proposed a new model to predict the SPAD DCR increase considering afterpulsing. The model gives a better fit to the experimental data of SPADs with high afterpulsing probability.

We theorize the observation of afterpulsing and the DCR level population in the megapixel SPAD camera is evidence of multiple trap energy levels created by radiation damage. It is thought that these energy levels represent the divacancies, oxide–vacancy complexes, and phosphorus–vacancy complexes created by proton radiation. We also discovered how the DCR level can be determined by different trap energy levels before irradiation, and it is more dependent on the number of defects created after irradiation. The DCR RTS effect also shows how proton radiation can create bi-stable or multi-stable defects. Depending on the spatial position of the defects in the silicon bulk, the trap energy level, and the trap density, released charges can worsen the overall DCR level or contribute to afterpulsing.

We observed the recovery of SPAD DCR during irradiation under room-temperature annealing and high-temperature annealing. The recovery during irradiation and under room temperature is likely evidence of the transient response of ionizing damage. Dark counts originating from the oxide–silicon interface are reduced as the oxide-trapped charges are neutralized. The SPAD DCR recovery follows a logarithmic trend similar to radiation effects on MOS devices. The DCR further improves after high-temperature annealing. The recovery trend follows the defect concentration versus annealing temperature of the phosphorus–vacancy complex, indicating the drop in DCR could be the result of phosphorus–vacancy annealing. The activation energy analysis after annealing also shows that there are remaining divacancies and phosphorus–vacancy complexes.

As the radiation damage is a fully probabilistic event, smaller SPADs show natural radiation hardness as their photocollector regions are less likely to be damaged. The observed stepwise increase in DCR agrees with the argument that DCR will not increase in the absence of defect creation within or near the photocollector region of the target SPAD. For applications requiring high radiation tolerance and low DCR, we conclude that a smaller SPAD size is preferable. Although smaller SPADs usually result in lower PDP [59,60], the drawback can be improved with optical enhancements such as microlenses.

From previous studies, we can see that photodiodes and CCDs are also sensitive to radiation damage [4,5]. The increase in the dark current of photodiodes and CCDs can range from 2 to 4 orders of magnitude under similar radiation doses. With the ability of photon counting, SPADs have much more of an advantage compared to the conventional sensors that are currently implemented in space-based systems in terms of the variety of applications. To conclude, the observed radiation effects on SPADs in this work show how SPADs not only have the potential to be implemented in long-term space missions where the proton flux, e.g., ~10 protons/cm2·s at 10 MeV for 400 km polar orbit mission [61], is several orders of magnitude lower than that used during our experiments (Table 1), but also can be used in high energy physics studies where high radiation doses are expected.

## Figures and Tables

**Figure 1 sensors-22-02919-f001:**
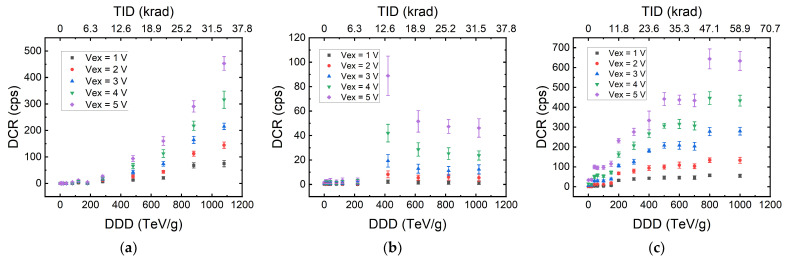
Three behaviors of SPAD DCR as observed in our experiments. The lower and upper axes represent the cumulative DDD and TID, respectively. Each 55 nm SPAD was characterized at 5 different Vex values over a cumulative dose; the SPAD DCR shows (**a**) continuous increase, (**b**) increase followed by decrease, and (**c**) stepwise increase. The three SPADs shown have active area radii of 2.43, 2, and 2 μm, respectively.

**Figure 2 sensors-22-02919-f002:**
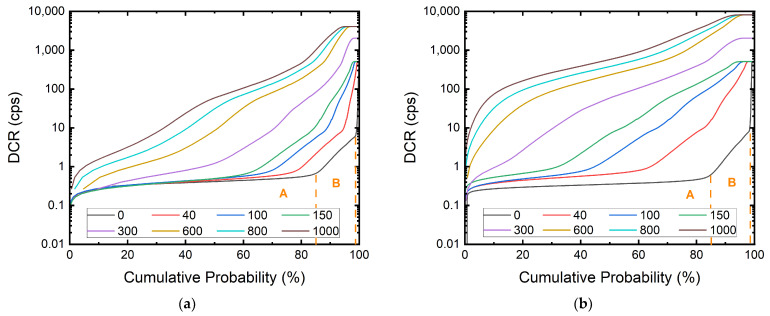
Cumulative DCR distribution of the 180 nm megapixel SPAD camera irradiated with (**a**) 100 MeV and (**b**) 10 MeV protons. The value in the legend represents the DDD in TeV/g; DDD = 0 represents the DCR before irradiation. The population at the plateau near 100% represents the saturated pixels. There are two clear knees at 85% and 99% cumulative probability before irradiation. The knees become less apparent at a higher cumulative dose.

**Figure 3 sensors-22-02919-f003:**
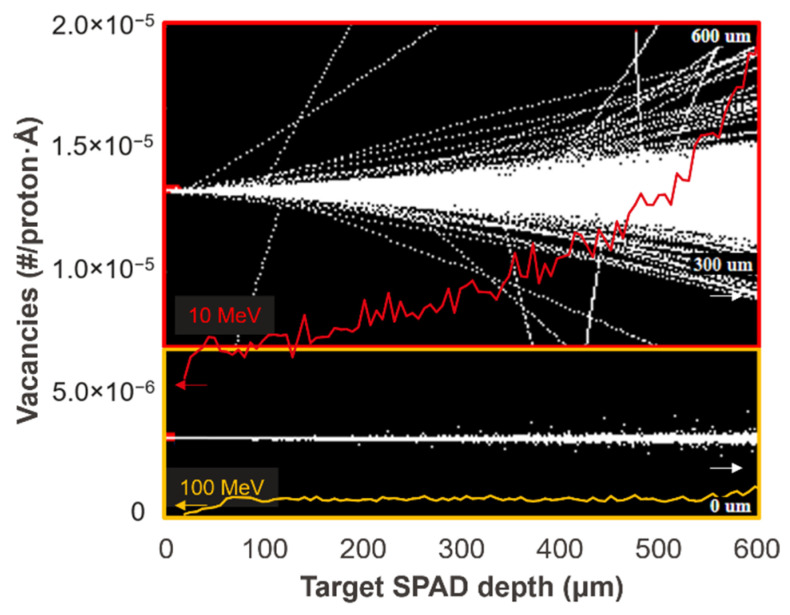
TRIM simulation result for 100 MeV and 10 MeV protons. The protons enter silicon from the left side, interact with silicon, and form the trajectories shown in white. The 10 MeV protons (top) interact with silicon more, while the 100 MeV protons (bottom) penetrate the silicon chip without much scattering. The orange and red curves represent the number of vacancies created per proton per angstrom in depth by 100 MeV and 10 MeV protons, respectively. The right *Y*-axis represents the lateral distribution of scattered protons.

**Figure 4 sensors-22-02919-f004:**
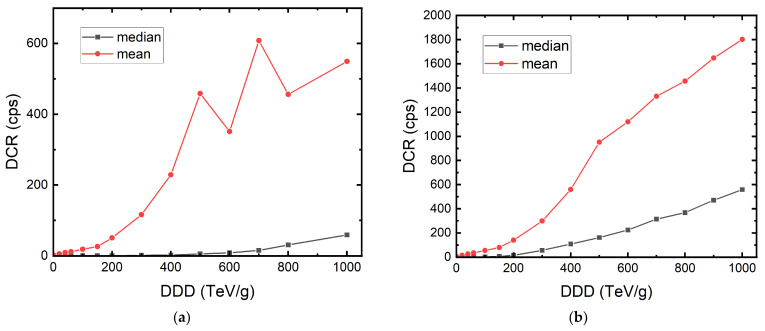
The mean and median DCR evolution of the 180 nm megapixel SPAD camera irradiated with (**a**) 100 MeV and (**b**) 10 MeV protons.

**Figure 5 sensors-22-02919-f005:**
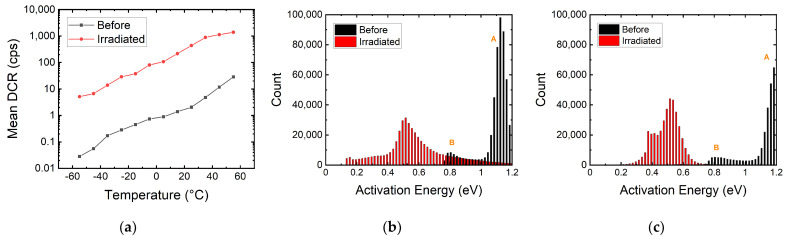
(**a**) Mean DCR of the 180 nm megapixel SPAD camera at different temperatures before and after irradiation. (**b**,**c**) Activation energy distribution of the megapixel SPAD camera before and after 100 MeV and 10 MeV proton irradiation. The shift in activation energy explains how trap energy can affect the DCR levels. Groups A and B correspond to the pixel groups in Figure 2.

**Figure 6 sensors-22-02919-f006:**
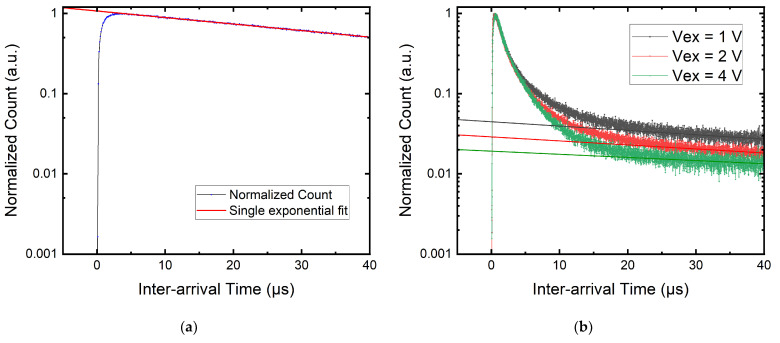
Afterpulsing measurement of a 55 nm SPAD with 2.43 μm active area radius: (**a**) no pulses lie above the Poisson single exponential fit, showing no afterpulse before radiation; (**b**) afterpulsing probability increases with increasing Vex after irradiation. The straight line represents the single exponential fit.

**Figure 7 sensors-22-02919-f007:**
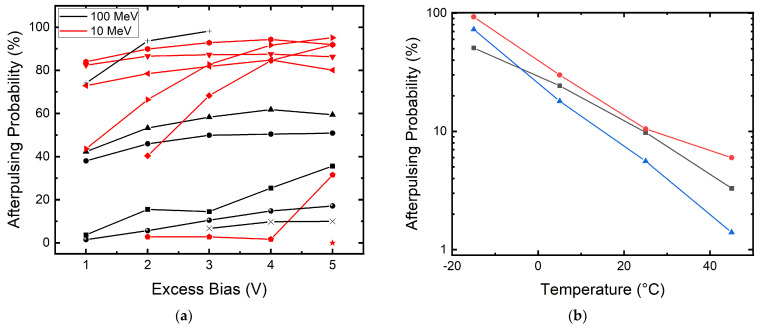
(**a**) APP at different Vex values of thirteen measured 55 nm SPADs. APP is higher at higher Vex values. Afterpulsing is more severe for SPADs irradiation with 10 MeV proton (shown in red). (**b**) APP at different temperatures of three measured SPADs (shown in three colors) irradiated with 100 MeV proton at Vex = 3 V. When lowering the SPAD operating temperature, the APP increases in an exponential trend.

**Figure 8 sensors-22-02919-f008:**
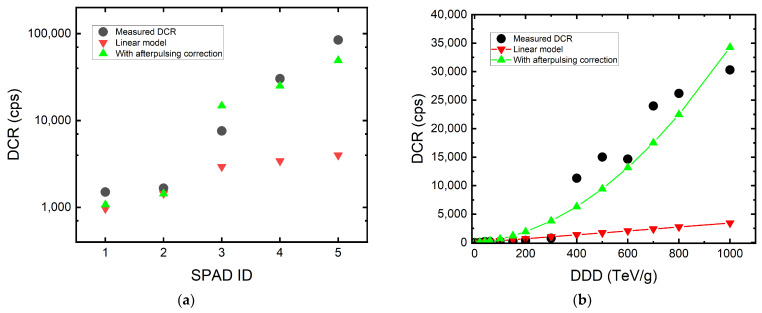
(**a**) The comparison between the linear model (red) and the proposed model (green). The proposed model gives a more accurate prediction for SPADs with a high DCR increase due to afterpulsing. (**b**) A prediction of DCR increases with cumulative dose using the proposed model (green) with an assumption of 90% of APP at 1 PeV/g.

**Figure 9 sensors-22-02919-f009:**
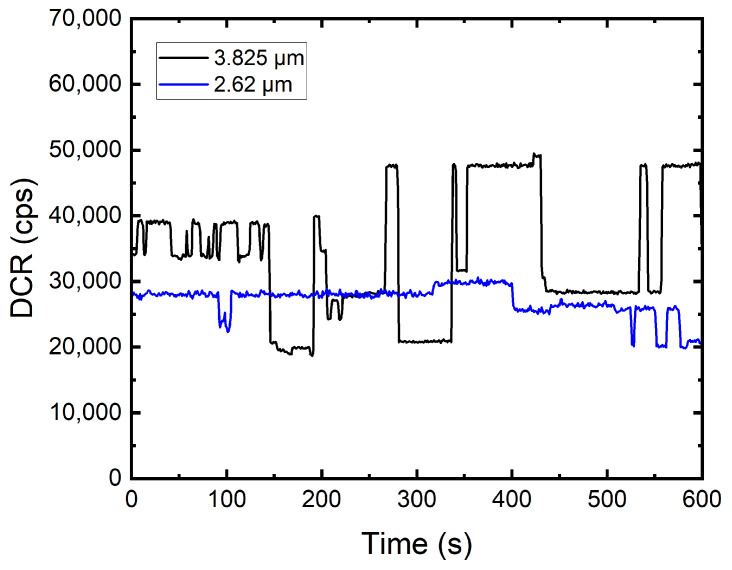
Two 55 nm SPADs showing DCR RTS at DDD of 80 TeV/g. The dark count at each second jumps between several DCR levels within a 600 s measurement. The black and blue curves represent SPADs with active area radii of 3.825 and 2.62 μm, respectively.

**Figure 10 sensors-22-02919-f010:**
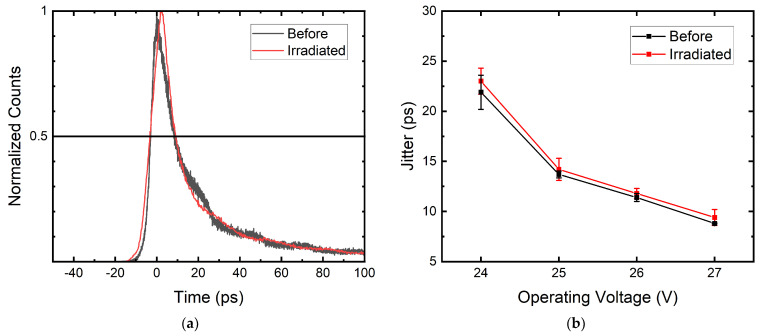
(**a**) Jitter measurement of the 180 nm accurate timing SPAD with 8.8 μm radius at 26 V operating voltage before and after radiation. (**b**) Jitter comparison at different operating voltages before and after radiation. The error bar represents the standard deviation between repeated measurements.

**Figure 11 sensors-22-02919-f011:**
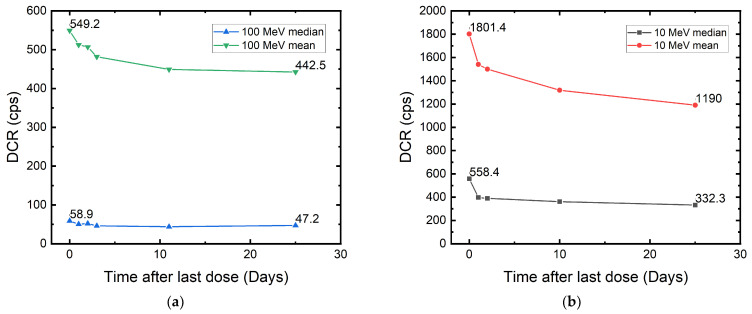
Evidence of room-temperature annealing of the 180 nm megapixel SPAD camera irradiated with (**a**) 100 MeV and (**b**) 10 MeV protons.

**Figure 12 sensors-22-02919-f012:**
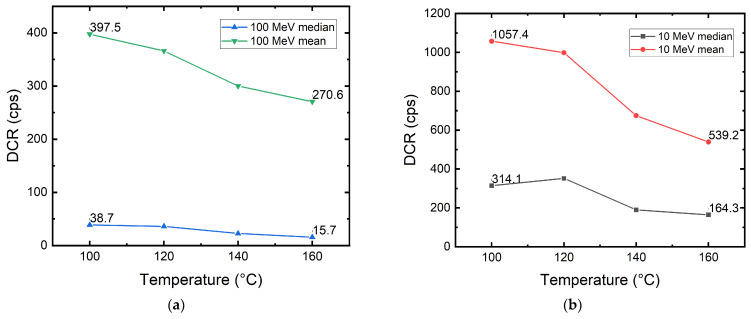
High-temperature annealing results of the 180 nm megapixel SPAD camera irradiated with (**a**) 100 MeV and (**b**) 10 MeV protons.

**Figure 13 sensors-22-02919-f013:**
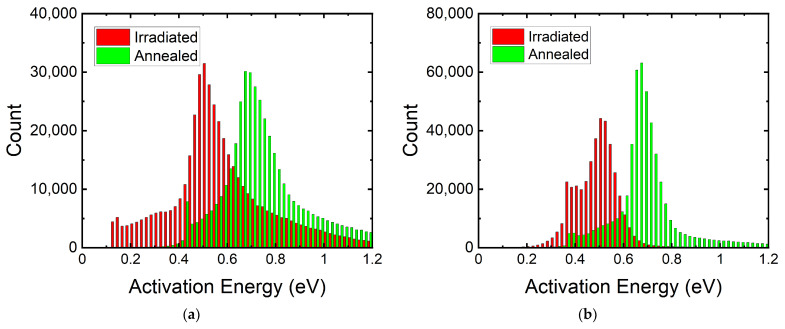
Activation energy distribution of the megapixel SPAD camera after (**a**) 100 MeV and (**b**) 10 MeV proton irradiation and after high-temperature annealing.

**Table 1 sensors-22-02919-t001:** Dose steps with target DDD and corresponding TID for 100 MeV and 10 MeV protons (* 100 TeV/g per step from step 9 to step 12).

		100 MeV			10 MeV		
Dose Steps	DDD (TeV/g)	TID (krad)	Flux (Proton/cm^2^/s)	Total Fluence (Proton/cm^2^)	TID (krad)	Flux (Proton/cm^2^/s)	Total Fluence (Proton/cm^2^)
0	0	0	0	0	0	0	0
1	20	0.63	1.6 × 10^7^	6.78 × 10^9^	1.18	1 × 10^7^	2.16 × 10^9^
2	40	1.26	1.6 × 10^7^	1.36 × 10^10^	2.36	1 × 10^7^	4.32 × 10^9^
3	60	1.89	1.6 × 10^7^	2.03 × 10^10^	3.83	1 × 10^7^	6.48 × 10^9^
4	100	3.15	3.0 × 10^7^	3.39 × 10^10^	5.89	1 × 10^7^	1.08 × 10^10^
5	150	4.72	3.0 × 10^7^	5.08 × 10^10^	8.83	1 × 10^7^	1.62 × 10^10^
6	200	6.30	3.0 × 10^7^	6.78 × 10^10^	11.8	1 × 10^7^	2.16 × 10^10^
7	300	9.45	6.0 × 10^7^	1.02 × 10^11^	17.7	1 × 10^7^	3.24 × 10^10^
8	400	12.6	6.0 × 10^7^	1.36 × 10^11^	23.6	1 × 10^7^	4.32 × 10^10^
9–12 *	…	…	1.0 × 10^8^	…	…	1 × 10^7^	…
13	900	28.3	1.0 × 10^8^	3.05 × 10^11^	53.0	1 × 10^7^	9.72 × 10^10^
14	1000	31.5	1.0 × 10^8^	3.39 × 10^11^	58.9	1 × 10^7^	1.08 × 10^11^

## Data Availability

Data available in a publicly accessible repository. The data presented in this study are openly available in Zenodo at 10.5072/zenodo.1051383.

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
