# Peer review of "Radiation Hardness Study of Single-Photon Avalanche Diode for Space and High Energy Physics Applications"

_sensors, 2022, doi:10.3390/s22082919_

Round 1
Reviewer 1 Report
Dear Authors,
This is an interesting study of the radiation hardness of SPAD devices. The measurement and irradiation methodologies are solid and broadly well performed, particularly the measurement of DCR immediately after irradiation to avoid self-annealing is well thought out. The results of the study would be of interest to researchers in the field.
However, this work is not very well cited with many references to researchers within their own group and few outside while not including appropriate references for similar studies on SPADs and CMOS image sensors. As perhaps a result of this, conclusions are made that are not supported by the results presented in this paper and at times the authors also demonstrate an incomplete device physics understanding that prevents valid conclusions from being reached. Please see the specific comments below for details.
This paper therefore requires major revisions prior to publication.
Specific comments for the Authors:
Abstract
The abstract is weak and reads like an introduction whereas it should summarize briefly all of the key points of the work the main including the findings and conclusions. The authors should refer to the Sensors author guidelines: https://www.mdpi.com/journal/sensors/instructions and rewrite the abstract accordingly. For example, the first two sentences of the current abstract should be in the introduction, not the abstract. A suggested first sentence of the abstract might read: “The operation of 55nm BCD and 180nm CMOS Single Photon Avalanche Diodes (SPADs) before and after irradiation with 10 and 100 MeV protons is studied and it is found that…” Many authors make similar mistakes writing abstracts and so this is not an unusual criticism of many published papers, but just because many authors write in this manner does not make it correct.
Introduction
The large number of self-references by the same authors in the first paragraph of the introduction (Refs. 1-11) does not inspire confidence that the authors have performed a thorough literature review to place their work in the context of the SPAD and radiation-impact-on-device fields. The reference list should be expanded to include more references with authors of no commonality to this work.
Materials & Methods
A cross sectional diagram for each of the SPAD structures irradiated should be included in order for the reader to understand how SPAD structural differences might impact the result. The lack of a SPAD structure diagram seriously impacts the understanding of this work. Additionally, the incident direction of the irradiation relative to the SPAD orientation should also be indicated.
Results
Breakdown voltage
What is the breakdown voltage of these devices? It is not listed.
I agree that irradiation should not change the BV unless amorphising-level doses are used. So I agree with the decision not to put these results in the paper to save space.
DCR
The results for the stepwise dark current are interesting, and I tend to agree with the hypothesis that this is caused by the probabilistic nature of damage creation. (note: this stepwise increase in DCR finding and associated hypothesis should be in the abstract!) However, the statement “To the best of our knowledge, this behavior has not been reported elsewhere to date.” is weakened by the lack of broad references in the introduction section, as mentioned previously. Moreover, this conclusion is also consistent with previous DCR studies that have found that only a single trap can cause a DCR increase, and the majority of SPADs have no traps.
Also, the English language and scientific nomenclature could be improved regarding beliefs, theories, and hypotheses. For example, line 140 might be better worded “We hypothesize that...” and on line 149 “This hypothesis is consistent with… ”
TRIM simulation results overlaid onto the SPAD structure should be provided. This enables the reader to understand the conclusion of why 10MeV protons cause a greater dark count increase than 100MeV protons. A visual explanation is always better than using words.
I also have serious issue and disagreement with the conclusions drawn from Figure 2 explained in lines 175 to 203. The explanation in this section needs to be linked much more closely with section 3.2.2 “Deep-level trap activation energy” and prior work from the literature on Dark Current Spectroscopy studies in CMOS image sensors, Dark Count Rate Spectroscopy in SPADs, and improved theoretical understanding of the SPAD structures under study (which means a diagram or even better – a TCAD 2D/3D field distribution plot). This means that 3.2.2 should follow immediately after this section (and be enhanced as described).
It is written that “The difference in DCR levels is likely to be caused by different types of defect within the silicon bulk or silicon-oxide interface.” But this cannot be entirely correct. Do your SPAD devices have the multiplication junction border onto the Si/SiO2 interface? Highly unlikely given that your DCR is so low. The explanation regarding silicon-oxide interface is therefore very unlikely, unless a supporting TCAD field plot can be provided. Additionally, regarding the first part of the explanation, different types of defects – this is possible, but the activation energy analysis in 3.2.2 suggests there are only 2 trap levels present after irradiation ~0.44 and ~0.55eV.
The authors also demonstrate their lack of understanding of device physics in regards to trap generated dark count by completely disregarding trap-assisted tunneling and the Poole-Frenkel effect from their analysis and conclusions. It is not just the trap type, but where the trap is created within the device that is important for the measured dark count rate and activation energy.
The authors should therefore expand their understanding of device physics and cite CMOS image sensor studies to broaden their understanding of trap-related mechanisms in silicon before revisiting their results and conclusions. Additionally, there have been several SPAD shrinking studies have revealed the same phenomenon of process-formed trap quantization in the dark count rate that may help the authors better understand their measurement results.
Section 3.2.2 Deep-level trap activation energy
There are no citations in this section indicating the authors have studied similar work by other authors and this shows in the analysis of the results. The experimental methodology appears valid, however. This section needs to be completely re-written in conjunction with 3.2 to include citations and learnings from similar spectroscopy studies in CMOS image sensors and SPADs.
For example, the authors write that “These two groups correspond to the two groups of SPAD shown in Figure 2, where the two knees are presented. This is proof of the correlation between the trap energy and the DCR level. Before irradiation, the DCR of the pixels is determined by the type of the existing defects from the fabrication process.” Which is only partially true given the results presented.
Yes, DCR correlates to Ea, but it also correlates to the magnitude of trap-assisted tunneling which is dependent on where the trap is created in the SPAD structure. Moreover, Ea cannot explain all of the variation in DCR since it is quite clear that irradiation produces ~0.44 and ~0.55eV defects. How can two trap levels produce continuously varying DCR according to SRH? Your results are not consistent with the conclusions drawn. It is also clear from the Fig 6-b and Fig 2-a that the majority of SPADs pre-irradiation do not have deep level traps in them. Please send my congratulations to the process integration team involved!
3.2.1. Afterpulsing probability
This is a thorough and strong study of afterpulsing and radiation dose’s impact on it. To this end Figure A1 should be integrated into this section and not in the appendix since it is directly referred to in the body text. The appendix should only have the abbreviations.
The only objection that I have in this section is the statement that “We believe this is further evidence that several trap energy levels are created during irradiation.” For similar reasons to my comments regarding dark current.
3.3. Random telegraph signal (RTS)
RTS alone is typically used to refer to readout noise in circuits. It would be therefore more clear to name this section Dark Count Rate Random Telegraph Signal since it appears that DCR observed popcorn noise behavior. Did it?
The authors should consider a figure showing this phenomenon.
This section is very short – why? Was DCR RTS observed in non-irradiated parts? It is not clear.
I agree that DCR RTS “can be explained by the multistable defect within the silicon bulk, resulting in a random change of two or more DCR levels.” But this statement contradicts the following sentence “This is another proof of the creation of multiple defect levels due to radiation” – was it one trap or two that were created? I would suspect 1 trap that was bi-stable, which is different from what the authors describe.
3.4. Photodetection probability
I agree that PDP should not be impacted by radiation, just like BV. However, no results are also shown. Perhaps the authors should explain the reason why the results were not shown was to save space and length?
Jitter
I agree that irradiation should not change the jitter but there are flaws in the experimental study that do not robustly support this conclusion. The before and after irradiation histograms in Fig 7a are slightly different which would suggest that radiation had some impact.
However, I know that Jitter is a very complex and time-consuming measurement to perform and many different factors can impact the result. The authors should have measured the device or multiple devices multiples times before and after irradiation to quantify their experimental error and variability. The authors write that “The slight difference between measurements can be caused by sample alignment with the experimental setup which creates an error at picosecond level.” Which I would tend to agree with, but the reader has no quantification of the authors experimental procedure. The difference before/after irradiation is just passed off as measurement variation without any evidence provided to support this conclusion.
To rectify this experimental shortcoming the authors should overlay the existing data with multiple jitter measurements of the same device on different days to show that the difference in results is down to variation and not irradiation. Failing this, the whole section needs to be removed because no conclusion based on the results can be drawn.
3.6.1. Room-temperature annealing
Figure 8 – array average or median DCR? Looks like median, but this should be clearly indicated in the axis label.
The authors write that “The median DCR recovered percentages are 20 % for the 100 MeV and 40 % for the 10 MeV DUT. This can be due to the higher TID received by the 10 MeV sample since displacement damages are less likely to recover under room temperature. ”
Or maybe it is simply because the 10MeV sample has more scope for “easy” improvement?
It would be more informative in this section to look at pixel-wise DCR recovery rather than array median. Information may be gleaned from whether low Ea pixels are likely to recover or not and how the DCR distribution changes after annealing would also be interesting. This section has lots of scope for improvement.
The authors write that:
“We believe this to be evidence of the transient response, caused by ionizing damage at the oxide interface. Due to cumulative TID, charges can be trapped at the surface of the SPAD, where oxide is used for passivation, or at the isolation-silicon interface, where oxide is used in trenches to prevent electric leakage between adjacent devices. The reduction in SPAD DCR can be originating from the neutralization of oxide-trapped charges. “
Without a SPAD structure and depletion profile indication it is impossible to know if this hypothesis is supported by the experiment results. It is likely from the DCR of pre-irradiation devices that the SPAD high field regions are well separated from the interfaces otherwise the DCR would not be as low as ~0.3cps.
3.6.2. High-temperature annealing
Figure 9 – array average or median DCR? Looks like median, but this should be clearly indicated in the axis label.
The result of decreasing DCR with increasing annealing temperature is reasonable.
However, the discussion in lines 372-381 are unsupported by the data presented. Studying the Ea before/after annealing is required to form any conclusions regarding trap annealing. Indeed, are the authors aware that the ~0.44eV defect they find in the activation energy study has been previously correlated with the phosphorus vacancy complex they mention in this section (although not in the citation provided)?
A plot of the median DCR is insufficient to draw these conclusions and instead what happened to the activation energy distribution after annealing at high temperature would be required. Alternatively, these unsupported speculations could be removed and replaced with the simple observation that DCR decreases after annealing and more at higher temperatures.
Discussion
This section should be reconsidered and re-written after the preceding comments have been addressed.
Author Response
The authors would like to thank the reviewer for the comments and suggestions. We have modified the abstract based on the review. For the point-by-point response to the reviewer's comments, please see the attachment.

Reviewer 2 Report
The paper is very original and presents the results clearly. I do have some minor remarks for improving the readability of the paper.
General remark: I see there are 3 SPAD types used. In my opinion, its better either to name them (for ex, DUT1,...) and/or mention them in the Figures. Some figure captions have no information on which SPAD while some do (Figure 6 says its a 180 nm SPAD). Please be more consistent.
Figure1: Its mentioned in L140 about the size of the SPADs. Please mention the size of the SPADs used.
Figure5: (a) Please make a legend inset that the red is with 10 MeV and 100 MeV. It could be easier for the reader.
Section 3.3: RTS data graph will be useful here. Also, Can you please confirm there was no RTS behavior before the irradiation.
Author Response

(The authors gave the same response as above.)

Reviewer 3 Report
The Authors report the study on the radiation hardness of single-photon avalanche diode, showing high tolerance to ionizing and displacement damage caused by protons. The results have been experimentally performed and result very interesting. I suggest the major revision of the manuscript. See the following comments:
- In the Introduction Section, the overview on the devices for Space applications should be enlarged, reporting the hardness of several devices (see “Radiation damage of electronic components in space environment,” Nucl. Instrum. Methods Phys. Res., Sect. A 514(1-3), 112–116 (2003); “Radiation sensitivity of light emitting diodes (LED), laser diodes (LD) and photodiodes (PD),” IEEE Trans. Nucl. Sci. 39(3), 423–427 (1992); “Radiation damage and annealing in 1310-nm InGaAsP/InP lasers for the CMS tracker,” In Photonics for Space Environments VII (International Society for Optics and Photonics, 2000), 4134, pp. 176–185, “Measured radiation effects on InGaAsP/InP ring resonators for space applications”,Optics Express, 27(17), 24434-24444 (2019), “Radiation hardness of high-Q silicon nitride microresonators for space compatible integrated optics,” Opt. Express 22(25), 30786–30794 (2014),), useful for Space applications. PDs are part of just a single category of devices suitable for Space. Moreover, the Authors should report more details on the Space environment, and its main risks on the mission success (see “The PAMELA experiment on satellite and its capability in cosmic rays measurements,” Nucl. Instrum. Methods Phys. Res., Sect. A 478(1-2), 114–118 (2002).)
- In order to help the reader to rate the proposed manuscript, the Authors should report the benefits of SPADs with respect to std PDs, also in the Space environment.
- The Authors report in the Section 1: “…For SPADs, it has been shown that dark count rate (DCR) increases due to displacement damage caused by protons, and ionizing damage caused by X-ray, alpha particles, and neutrons [12-14]. In this paper, we study the effect of proton, or hydrogen nuclei radiation, which is one of the main radiation sources given by cosmic rays, on SPADs.”. It is not clear from these sentences the improvement with respect the state of the art.
- The design and device description miss in the manuscript. The Authors should describe the DUT, also reporting a sketch. Moreover, fabrication details should be reported. The geometrical features should be justified by modelling.
- The Eq. (3) represents an adding value of the manuscript. More details should be inserted, reporting also how the equation has been conceived.
- To increase the manuscript attractiveness, the Authors could also report the expected performance of the device under proton irradiation. The Single Event Effect should be also discussed.
- About the manuscript style:
- Insert the acronyms in the abstract.
- Line 288, insert Reference.
- Singular/plural on line 299.
- Report the legend in Fig. 2.
- Insert unit of measure in all equations.
Author Response

(The authors gave the same response as above.)

Round 2
Reviewer 1 Report
Dear Authors,
Thank you for addressing my comments. As a result the paper is much improved and worthy of publication in Sensors.
The only minor comment I have is the use of "believe" and "belief" which are typically used in a religious, not scientific, context. Phrases such as "It is hypothesised that" and "It is thought that" "It is theorised that" are more scientific and more appropriate for scientific publication.
Good luck with the thesis.
Author Response
The authors would like to thank the reviewer for the positive feedback. We have modified the manuscript based on the comment.
Reviewer 3 Report
The Authors have replied to the Reviewers’ comments with satisfactory responses. The new version of the manuscript has been modified following the Reviewers’ suggestions. Therefore, I recommend the publication of the manuscript as it is.
Author Response
The authors would like to thank the reviewer for the positive feedback.